# SeaMAE: Masked Pre-Training with Meteorological Satellite Imagery for Sea Fog Detection

Haotian Yan [1,†], Sundingkai Su [1,†], Ming Wu [1], Mengqiu Xu [1], Yihao Zuo [1], Chuang Zhang [1] and Bin Huang [2,*]

1  Artificial Intelligence School, Beijing University of Posts and Telecommunications, Beijing 100876, China; yanhaotian@bupt.edu.cn (H.Y.); susundingkai@bupt.edu.cn (S.S.); wuming@bupt.edu.cn (M.W.); xumengqiu@bupt.edu.cn (M.X.); zuoyihao@bupt.edu.cn (Y.Z.); zhangchuang@bupt.edu.cn (C.Z.)
2  National Meteorological Center, China Meteorological Administration, Beijing 100081, China
*  Correspondence: huangb@cma.gov.cn
†  These authors contributed equally to this work.

**Abstract:** Sea fog detection (SFD) presents a significant challenge in the field of intelligent Earth observation, particularly in analyzing meteorological satellite imagery. Akin to various vision tasks, ImageNet pre-training is commonly used for pre-training SFD. However, in the context of multi-spectral meteorological satellite imagery, the initial step of deep learning has received limited attention. Recently, pre-training with Very High-Resolution (VHR) satellite imagery has gained increased popularity in remote-sensing vision tasks, showing the potential to replace ImageNet pre-training. However, it is worth noting that the meteorological satellite imagery applied in SFD, despite being an application of computer vision in remote sensing, differs greatly from VHR satellite imagery. To address the limitation of pre-training for SFD, this paper introduces a novel deep-learning paradigm to the meteorological domain driven by Masked Image Modeling (MIM). Our research reveals two key insights: (1) Pre-training with meteorological satellite imagery yields superior SFD performance compared to pre-training with nature imagery and VHR satellite imagery. (2) Incorporating the architectural characteristics of SFD models into a vanilla masked autoencoder (MAE) can augment the effectiveness of meteorological pre-training. To facilitate this research, we curate a pre-training dataset comprising 514,655 temporal multi-spectral meteorological satellite images, covering the Bohai Sea and Yellow Sea regions, which have the most sea fog occurrence. The longitude ranges from 115.00°E to 128.75°E, and the latitude ranges from 27.60°N to 41.35°N. Moreover, we introduce SeaMAE, a novel MAE that utilizes a Vision Transformer as the encoder and a convolutional hierarchical decoder, to learn meteorological representations. SeaMAE is pre-trained on this dataset and fine-tuned for SFD, resulting in state-of-the-art performance. For instance, using the ViT-Base as the backbone, SeaMAE pre-training which achieves 64.18% surpasses from-scratch learning, natural imagery pre-training, and VRH satellite imagery pre-training by 5.53%, 2.49%, and 2.21%, respectively, in terms of Intersection over Union of SFD.

**Keywords:** sea fog detection; pre-training; masked autoencoders; meteorological satellite imagery

## 1. Introduction

The precise detection of sea fog (SFD) using meteorological satellite imagery holds substantial implications for a wide range of marine activities. Meteorological bureaus continuously monitor their jurisdictional ocean areas to forecast weather, manage fishing and shipping, and relieve meteorological disasters (Figure 1a) [1,2]. In this study, our focus lies on the Bohai Sea and Yellow Sea regions (Figure 2a), utilizing deep-learning techniques to tackle this problem.

Through an exhaustive review of previous studies on SFD [3–14], we observe that the question of pre-training SFD networks remains an open challenge. Some papers rely on ImageNet pre-training weights [3,4,6,8,10,14], while some others even opt to train SFD

models from scratch on fine-tuning data [5,7–9,11–13]. Lately, with the advancements in self-supervised learning [15–18], which is independent of expensive semantically labeled data, customizing pre-trained models on large-scale raw data has become a prevailing practice, especially in some specialized domains such as medical imagery [19,20] and VHR satellite imagery [21,22]. The existing VHR satellite imagery pre-training model may be deemed as an alternative suitable for SFD since the data in pre-training and fine-tuning are both captured by satellites from aerial angles. However, vast distinctions between the VHR and meteorological satellite imagery, such as the number and types of bands, image resolution, and observation targets, necessitate the exploration of meteorological pre-training for SFD.

This paper leverages the rich data obtained from the Himawari-8 satellite as the foundation for our investigation [23]. Indeed, the Himawari-8 satellite has emerged as a crucial contributor to the rapid progress in SFD. It offers the capability to capture up to 16 Advanced Himawari Imager (AHI) observation bands (Figure 2b) and provides substantial temporal publicly available data (Figure 3c). Hence, we curate a dataset comprising 514,655 samples as the pre-training data. To better differentiate the samples of our meteorological dataset from the counterparts, Figure 3a,b exhibits natural imagery of ImageNet [24] and VHR satellite imagery of Million-AID [22]. Based on these comparisons, it becomes evident that there exist significant domain gaps between meteorological satellite imagery versus both natural imagery and VHR satellite imagery. It is justifiable to argue that pre-training SFD models using meteorological satellite imagery can result in the acquisition of more valuable meteorological representations.

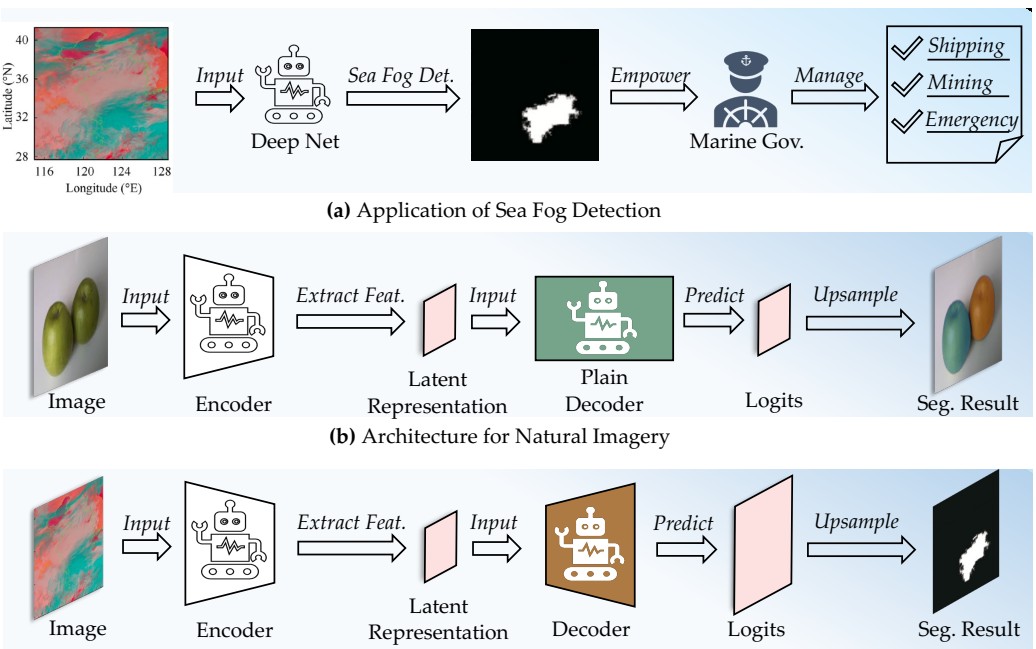

(a) Application of Sea Fog Detection

(b) Architecture for Natural Imagery

(c) Architecture for Meteorological Satellite Imagery

**Figure 1.** (**a**) illustrates how sea fog detection (SFD) contributes to marine management. (**b**) An effective network pipeline on segmenting natural imagery. The upsampling operation is intensive because it interpolates the logits with very large ratios (e.g., 8 or 16) to match the spatial size of the input. (**c**) A canonical network pipeline for SFD. The most apparent difference lies in the decoder part, where the latent representation is upsampled progressively, and the output layer upsample the logits very slightly, with a ratio of 2.

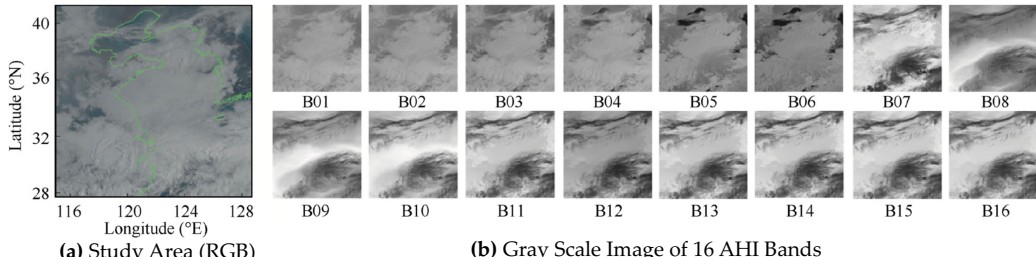

**(a)** Study Area (RGB)      **(b)** Gray Scale Image of 16 AHI Bands

**Figure 2.** (**a**) Study area. The green curve depicts the coastline. (**b**) Visualization of 16 AHI bands in grayscale.

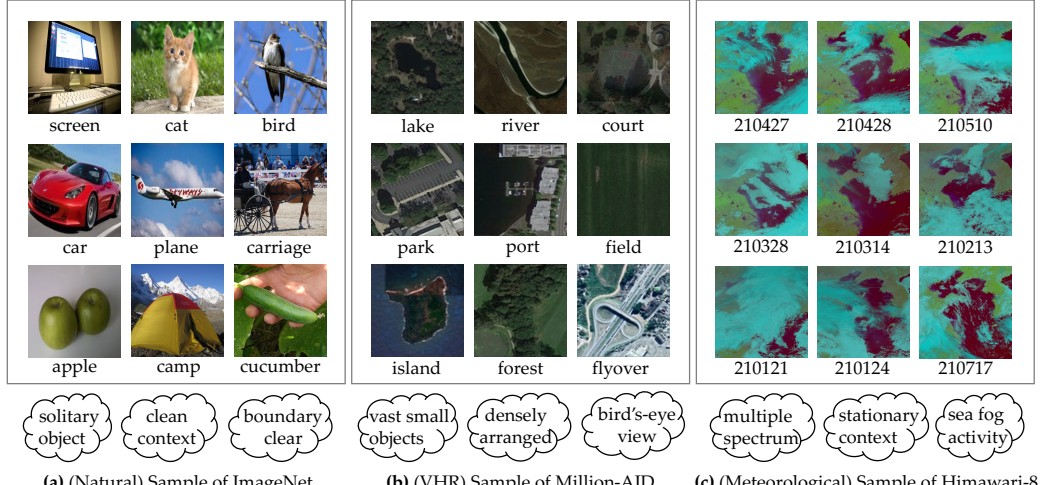

**(a)** (Natural) Sample of ImageNet     **(b)** (VHR) Sample of Million-AID     **(c)** (Meteorological) Sample of Himawari-8

**Figure 3.** (**a**) Samples from ImageNet dataset. (**b**) Samples from Million-AID dataset. (**c**) Samples from Himarwari-8. These samples are synthesized with (B03, B04, B14). The number under every image is the time when it is taken, in a format YYMMDD.

Besides the large-scale meteorological satellite imagery, another pivotal aspect of successful meteorological pre-training lies in the adoption of learning methodologies. Our approach builds upon Masked Image Modeling (MIM), as Masked Autoencoders (MAE) [15], a canonical method of MIM, have been empowering various realms through competently pre-trained Vision Transformers [25]. Given the absence of SFD networks incorporating a Vision Transformer as the backbone, we assert that employing MAE for pre-training can simultaneously fill this gap, making contributions not only to pre-training SFD networks but also to fine-tuning SFD networks.

Furthermore, we tweak the architecture of MAE by replacing its plain decoder with a convolutional hierarchical decoder [26,27]. The motivation stems from the performance of SFD improving a lot when the plain decoder in the SFD network is substituted by a hierarchical decoder [6,10]. The plain decoder indicates the upsampling is intensive [15,28], but the hierarchical decoder indicates the upsampling is progressive [29–31], as shown in Figure 1b,c. Regarding meteorological satellite imagery, the main objects it covers, such as clouds and fog, possess diverse and complex forms, along with intricate texture features and fragile edge information. Therefore, iterative upsampling operations are necessary for processing the latent representations to progressively restore spatial dimensions. Such an architectural idea can be defined as an inductive bias highly associated with meteorological satellite imagery, irrespective of the task being SFD or MIM. Due to the introduction of this inductive bias of the SFD network, we term our tweaked MAE as SeaMAE.

To be brief and clear, we enumerate the contributions of this work:

- This paper proposes meteorological pre-training for sea fog detection (SFD). To this end, we collect 514,655 Himawari-8 satellite multi-channel images and use MAE to learn representations from them. On SFD, this pre-training paradigm outper-

forms from-scratch learning, ImageNet pre-training, and VHR satellite imagery pre-training. The power of available large-scale meteorological satellite raw data is unparalleledly utilized.

- We investigate a decoder architecture tailored for meteorological pre-training, which results in a novel variant of MAE, SeaMAE. Specifically, an off-shelf decoder effective in SFD is utilized to model masked patches in pre-training. Meteorological pre-training on the proposed SeaMAE facilitates additional performance gains for SFD. To our knowledge, this paper first pioneers the application of Vision Transformers for SFD in this community.

- SeaMAE intrinsically introduces an architecturally end-to-end pre-training paradigm, wherein the decoders in both pre-training and fine-tuning share the same architecture, revolutionizing the previous routine that only the encoder of SFD is pre-trained. We manifest that the pre-trained decoder performs better than the from-scratch learning decoder on fine-tuning data. The extension of pre-trained components in the SFD network shows great promise.

- Finally, training SFD networks typically involves using either all bands or (3, 4, 14) bands. Generally, the former performs better than the latter. Our proposed learning paradigm can adapt to both training settings, and make the latter performance on par with the former because our proposed learning paradigm has learned representations from all bands during the pre-training.

## 2. Materials and Methods

This section contains three parts. The first part introduces the pre-training dataset as well as the collection thereof, and the fine-tuning dataset on which every learning paradigm will be verified. The second part elaborates on our new MAE with a decoder architecturally consistent with that for sea fog detection (SFD). In the last part, we comprehensively describe the pipeline of the proposed meteorological learning paradigm based on SeaMAE for SFD.

### 2.1. Himawari-8 Meteorological Satellite Imagery

Himawari-8 is a geostationary weather satellite operated by the Japan Meteorological Agency (JMA) [23]. It was launched in 2014 and is positioned over the western Pacific Ocean, providing continuous observation of the Asia-Pacific region. Himawari-8 captures high-resolution imagery and monitors weather patterns, cloud cover, and other atmospheric states. It has been instrumental in enhancing weather forecasting and disaster management efforts in the region. We use its meteorological satellite imagery to construct the pre-training dataset for self-supervised learning, and the fine-tuning dataset for SFD.

#### 2.1.1. Pre-Training Dataset

The meteorological satellite imagery in the area of Bohai Sea and Yellow Sea (115.00°E to 128.75°E, 27.60°N to 41.35°N) from the year 2018 to 2021 serves as our database. To accomplish the meteorological pre-training, we collect all 2018 images within the study region, regardless of whether these images capture any sea fog activity, and crop them with a spatial size of $256 \times 256$. As a result, the pre-training dataset involves 514,655 samples. Despite the meteorological dataset not being dominant in terms of quantity compared to ImageNet (1.43 million) or Million-AID (1.1 million), it possesses 16 channels and every band has special meteorological, cloud, and marine information, as reported in Table 1 and visualized in Figure 2, which can provide the SFD model more meteorological representations in the ocean region. Furthermore, its coverage area aligns precisely with the scope of SFD in that, generally speaking, SFD is also a geostationary interpretation task on satellite imagery, distinguishing it from numerous other satellite imagery pre-training datasets.

**Table 1.** Information of AHI observation bands.

| No. | Name | Type | Detection Category |
|---|---|---|---|
| B01 | V1 | Visible | Vegetation, aerosol |
| B02 | V2 | Visible | Vegetation, aerosol |
| B03 | VS | Visible | Low cloud (fog) |
| B04 | N1 | Near Infrared | Vegetation, aerosol |
| B05 | N2 | Near Infrared | Cloud phase recognition |
| B06 | N3 | Near Infrared | Cloud droplet effective radius |
| B07 | I4 | Infrared | Low cloud (fog), natural disaster |
| B08 | WV | Infrared | Water vapor density from troposphere to mesosphere |
| B09 | W2 | Infrared | Water vapor density in the mesosphere |
| B10 | W3 | Infrared | Water vapor density in the mesosphere |
| B11 | MI | Infrared | Cloud phase discrimination, sulfur dioxide |
| B12 | O3 | Infrared | Ozone content |
| B13 | IR | Infrared | Cloud image, cloud top |
| B14 | L2 | Infrared | Cloud image, sea surface temperature |
| B15 | I2 | Infrared | Cloud image, sea surface temperature |
| B16 | CO | Infrared | Cloud height |

### 2.1.2. Fine-Tuning Dataset

The fine-tuning dataset comprises a training set and a validation set. The training set includes images from the period of 2019 to 2020, while the validation set consists of images from 2021. Unlike the pre-training dataset that involves all samples of the entire year, fine-tuning focuses merely on the images that capture sea fog.

We organize a group with meteorological expertise in recognizing sea fog from the meteorological satellite imagery, employing them to annotate the sea fog of training and validation set in pixels. Consequently, a pixel-level annotated SFD dataset is contributed to the community by us, including 1128 training image-label pairs and 382 test image-label pairs involving 108 sea fog events, both of which are with spatial sizes of (1024, 1024) covering the ocean region with the longitude from 115.00°E to 128.75°E as well as the latitude from 27.60°N to 41.35°N. We believe this dataset will be of great applicability for researchers in this field due to the delicate annotation, sufficient samples, and various sea fog activities. Therefore, another article will be organized to provide a deeper introduction and a wider analysis of this SFD dataset, and the dataset will also be publicly available soon. Before that, we report some state-of-the-art (SOTA) methods' performance on this fine-tuning dataset to quantify its challenge in advance, as displayed in Table 2.

**Table 2.** Comparison of SOTA methods on our fine-tuning dataset. The bold indicates the highest result.

| Backbone | Methods | IoU | Acc |
|---|---|---|---|
| CNN | Deeplabv3+ [28] | 56.10 | 68.89 |
| | Scselinknet [3] | 55.29 | 68.18 |
| | UNet [31] | **58.44** | **70.23** |
| | UNet++ [29] | 56.95 | 69.30 |
| | Attetion-unet [30] | 56.35 | 68.94 |

### 2.2. SeaMAE

Similar to the majority of Masked Image Modeling (MIM) methods, our approach utilizes the masked autoencoder (MAE) to learn visual representations [15], as illustrated in Figure 4. In line with a standard MAE, the input image in our SeaMAE model is divided into non-overlapping patches, with 75% of the patches being randomly masked. The encoder, which is a plain Vision Transformer, operates solely on the unmasked patches and produces latent representations. The decoder takes these latent representations along with learnable mask tokens as inputs and reconstructs the masked patches at the pixel level.

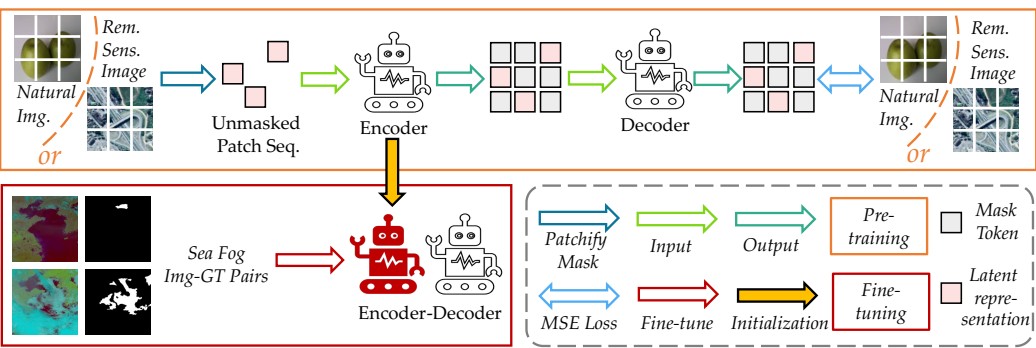

**Figure 4.** Illustration of the MIM pre-training and fine-tuning in existing sea fog detection applications.

### 2.2.1. SFD Driven by ViT

Adopting MAE to pre-train means that Vision Transformer is taken as the backbone of the network for sea fog detection. Since there have been never previous works probing this topic, we must first introduce a simple method to utilize Vision Transformer, particularly ViT [25]. Now that the hourglass architecture, or so-called U-shape architecture, has proved effective in SFD [26,31], the key issue is how to extract the feature-map hierarchy from ViT in that the U-shape network transfers the encoder's feature to the decoder in a stage-to-stage fashion by building skip connections. Unlike hierarchical networks [32] which can naturally generate feature hierarchy, a common manner to handle the plain ViT is separating the ViT into four stages regularly [33], hooking the last feature-map of each stage, and rescaling them to multiple levels.

We propose another easier method to deal with this. Specifically, only the last feature map is used to generate all-level feature maps because it has the strongest representations. By denoting $F$ as the feature-map output by ViT, the generation of the feature-map corresponding $n$-th stage on the top of the decoder can be formulated by:

$$F_n = Bicubic(F, ratio = 2^{3-n}), n \in [1, 4], \tag{1}$$

where *Bicubic* is a function acting on $F$ through bicubic interpolation with a re-sampling *ratio*. Generally, ViT's output feature map has an output stride of 16 compared to the input 2D image, hence the values of *ratio* are 0.5, 1, 2, and 4. Given such a group of outputs, most decoders for sea fog detection can be used without any bells and whistles.

### 2.2.2. Hierarchical Decoder

As aforementioned in Section 1, the downstream task of SFD motivates the utilization of a hierarchical decoder instead of a plain decoder in MIM pre-training. In terms of effortlessness, we select LinkNet [26], a widely favored hierarchical decoder in SFD [3,6], to serve as the decoder during both the pre-training and fine-tuning stages. The decoder has 4 blocks and the $n$-th decoder block takes its corresponding feature $F_k$ of the feature pyramid as input. The structure of one decoder block is visualized in Figure 5c. It upsamples $F_n$ and outputs $E_n$ which is added to $F_{n-1}$, and the summation is input to the $(n-1)$-th decoder block.

### 2.2.3. Skip Connection for Masked Input

The skip connection plays a crucial role in the U-shape network, connecting the backbone and the decoder. Although implementing the skip connection is straightforward during fine-tuning, it becomes more challenging in MIM pre-training, where the patch sequence in the backbone is masked. For the masked patch sequence, directly using interpolation to upsample it may associate with potential risks due to the considerable information loss. We hence perform interpolation on the concatenation of masked patches and mask tokens, as shown in Figure 5b. The operation of interpolation is identical to Equation (1).

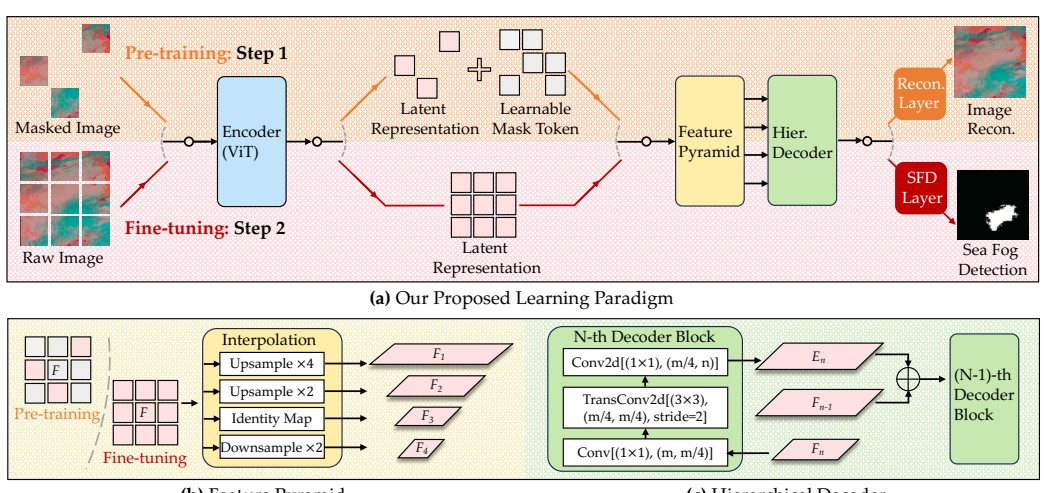

**(a)** Our Proposed Learning Paradigm

**(b)** Feature Pyramid

**(c)** Hierarchical Decoder

**Figure 5.** (**a**) illustrates our proposed learning paradigm. (**b**) details our method to output the feature pyramid with only the last feature from ViT. (**c**) illustrates the hierarchical decoder architecture and components of one decoder block.

### 2.3. Learning Paradigm

Like the conventional "first pre-training then fine-tuning" learning paradigm illustrated in Figure 4, our resulting learning paradigm also contains two steps, as shown in Figure 5a. However, different from the steps of the previous learning paradigm shown in Figure 4 which are conducted on two different architectures with imagery from different domains, our method's two steps depend on the same architecture except for the output layer. The first step is pre-training when SeaMAE learns representations on 514,655 meteorological satellite images. By denoting the output of hidden layers as $F_h \in \mathbb{R}^{H*W*dim}$, the output layer can be defined as $W_{rec} \in \mathbb{R}^{dim*3}$, and the operation of the output reconstruction layer can be formulated by

$$O_{rec} \in \mathbb{R}^{H*W*3} = F_h \times W_{rec}. \tag{2}$$

The objective function is Mean Square Error (MSE):

$$MSE = \frac{1}{n} \sum_{i}^{n} (y_i - \hat{y}_i)^2, \tag{3}$$

where $n$ is the number of masked patch tokens, $y_i \in \mathbb{R}^{p*p*3}$ is $i$-th masked patch token of raw image, and $\hat{y}_i \in \mathbb{R}^{p*p*3}$ is $i$-th reconstructed counterpart ($p$ denotes the patch size) of $O_{rec}$.

In the second fine-tuning step, the SFD model driven by ViT is trained on labeled meteorological satellite imagery. For $F_h \in \mathbb{R}^{H*W*dim}$, the output layer can be defined as $W_{sfd} \in \mathbb{R}^{dim*1}$, and the operation of the output SFD layer can be formulated by

$$O_{sfd} \in \mathbb{R}^{H*W*1} = F_h \times W_{sfd}. \tag{4}$$

The objective function is binary cross-entropy (BCE) loss with sigmoid activating logits:

$$BCE = -\frac{1}{n}\sum_{i}^{n}(t_i \cdot log(o_i) + (1 - t_i) \cdot log(1 - o_i)), \tag{5}$$

where $n$ is number of pixels in the image, $t \in \{0, 1\}$ represents the background or sea fog, and $o$ is the sigmoid activating logits.

In summary, the pre-training is indeed architecturally close to the fine-tuning, as shown in Figure 5a. The distinctions between them only include mask tokens which are indispensable in pre-training but redundant in fine-tuning and the output layer whose output channel numbers are 3 or 1 in pre-training and fine-tuning as evident in Equation (2) and Equation (4), respectively. Therefore, two knowledge-transferring ways, i.e., loading pre-training weights, are proposed to initialize the parameters of the SFD network. (1) Only the knowledge of the SeaMAE encoder is transferred. (2) The entire knowledge of SeaMAE is transferred except for the mask tokens and output layer.

## 3. Results

In this section, we primarily validate the following hypotheses: (1) Under the standard MAE architecture, pre-training with meteorological satellite imagery yields better representations for sea fog detection (SFD) compared to pre-training with natural imagery or Very High-Resolution (VHR) satellite imagery. (2) The decoder of the pre-training MAE using meteorological satellite imagery should exhibit architectural characteristics that align with the SFD network. (3) In the context of sea fog detection, an architecturally end-to-end transfer of pre-trained knowledge surpasses the knowledge transfer limited to the backbone network alone. (4) Our proposed approach, SeaMAE, enjoys several desirable properties, such as data scalability, model-capacity scalability, fast pre-training convergence, and fine-tuning band adaptability. (5) Such architectural improvements result in a novel SeaMAE that performs better than existing state-of-the-art satellite imagery pre-training methods in terms of pre-training for SFD.

### 3.1. Implementation and Metric

Our hardware environment is a server with 4 NVIDIA3090 GPUS. We use PyTorch as the deep-learning framework to conduct all experiments. For the pre-training setting, we follow the learning protocols of the standard MAE [15]. However, towards the fast tempo of attaining results, all our pre-training experiments set learning epoch 300 instead of 800. For fine-tuning settings, we follow the learning protocols of [6]. In the training stage of fine-tuning, crops of $512 \times 512$ are input to the network, and in the evaluation stage, the whole image with a size of $1024 \times 1024$ is input to the network. The performance of detecting sea fog is evaluated through two metrics, sea fog Intersection over Union (IoU) and sea fog Accuracy (Acc.). The sea fog IoU can be calculated as:

$$IoU_{SF} = \frac{TP_{SF}}{TP_{SF} + FN_{SF} + FP_{SF}} \tag{6}$$

where $SF$ denotes sea fog, and $TP$, $FN$, $FP$ are True Positive, False Negative, and False Positive of detecting sea fog, respectively.

### 3.2. MAE Pre-Training on Meteorological Satellite Imagery

Table 3 compares pre-training on natural imagery—ImageNet [24]—and VHR satellite imagery—Million-AID [22]—and our collected Himawari-8 dataset. The experiment is conducted on three ViT backbones with different model capacities, ViT-Tiny (Ti), ViT-Small (S), and ViT-Base (B) [25]. First, Table 3 showcases the performance of from-scratch UNet, also displayed in Table 2, to define a baseline. For any variant of ViT, the performance of SFD with pre-trained models on any dataset outperforms the performance of from-scratch SFD by large margins. Except for ViT-Ti, pre-training with Million-AID results

in superiority to the ImageNet pre-training model on both ViT-S and ViT-B. On all three backbones, the meteorological pre-training outperforms natural imagery pre-training and VHR satellite imagery pre-training by large margins.

**Table 3.** Comparison of meteorological pre-training with ImageNet pre-training and from-scratch initialization on ViT-Ti, ViT-S, and ViT-B. Green numbers are the performance gains derived from changing pre-training data from VHR satellite imagery to meteorological satellite imagery.

| Pre-Train Method | Pre-Train Data | Encoder | Fine-Tune Decoder | IoU | Acc. |
|---|---|---|---|---|---|
| Supervised | ImageNet | CNN | UNet | 58.44 | 70.23 |
| From Scratch | — | ViT-Ti | LinkNet | 55.05 | 66.48 |
| MAE | ImageNet | ViT-Ti | LinkNet | 57.79 | 69.78 |
| MAE | Million-AID | ViT-Ti | LinkNet | 57.56 | 69.63 |
| MAE | Himawari-8 | ViT-Ti | LinkNet | 58.63 (+1.07) | 70.47 (+0.84) |
| From Scratch | — | ViT-S | LinkNet | 56.30 | 68.04 |
| MAE | ImageNet | ViT-S | LinkNet | 59.42 | 70.82 |
| MAE | Million-AID | ViT-S | LinkNet | 59.62 | 71.35 |
| MAE | Himawari-8 | ViT-S | LinkNet | 60.48 (+0.86) | 71.87 (+0.52) |
| From Scratch | — | ViT-B | LinkNet | 58.88 | 70.49 |
| MAE | ImageNet | ViT-B | LinkNet | 61.69 | 72.53 |
| MAE | Million-AID | ViT-B | LinkNet | 61.87 | 72.79 |
| MAE | Himawari-8 | ViT-B | LinkNet | 62.92 (+1.05) | 73.91 (+1.12) |

### 3.3. SeaMAE Pre-Training on Meteorological Satellite Imagery

Table 4 analyzes the introduction of SFD's architectural characteristics to the meteorological MIM pre-training, based on SeaMAE instead of MAE as the pre-training model. Also, the effectiveness is evaluated on three backbones. According to Table 4, the experimental results are divided into three groups based on different backbones, with the first two results in each group directly comparing the fine-tuning performance of SeaMAE and MAE on meteorological image pre-training. It can be observed that SeaMAE consistently outperforms MAE. As described in Section 2.2, the introduced architectural characteristics are hierarchical decoder and skip connection. Considering that the hierarchical decoder involves more tensor operations compared to the skip connection, Table 4 further probes the impact of retaining only the hierarchical decoder, i.e., removing the skip connection, in the SeaMAE on the fine-tuning performance. It is evident that the skip connection in the U-shape network under MIM learning, where the network takes masked images as input, cannot be overlooked.

**Table 4.** Comparison of meteorological pre-training based on MAE with SeaMAE. Green numbers are the performance gains derived from using SeaMAE instead of MAE. Red numbers are the performance drops derived from removing skip connections. ✓ means the skip connection is equipped. ✗ means the skip connection is removed.

| Pre-Train Method | Pre-Train Data | Encoder | Skip Connection | IoU | Acc. |
|---|---|---|---|---|---|
| MAE | Himawari-8 | ViT-Ti | – | 58.63 | 70.47 |
| SeaMAE | Himawari-8 | ViT-Ti | ✓ | 59.37 (+0.74) | 70.97 (+0.50) |
| SeaMAE | Himawari-8 | ViT-Ti | ✗ | 59.09 (−0.28) | 70.71 (−0.28) |
| MAE | Himawari-8 | ViT-S | – | 60.48 | 71.87 |
| SeaMAE | Himawari-8 | ViT-S | ✓ | 61.55 (+1.07) | 72.52 (+0.95) |
| SeaMAE | Himawari-8 | ViT-S | ✗ | 61.12 (−0.43) | 72.24 (−0.28) |
| MAE | Himawari-8 | ViT-B | – | 62.92 | 73.91 |
| SeaMAE | Himawari-8 | ViT-B | ✓ | 63.74 (+0.82) | 75.12 (+1.21) |
| SeaMAE | Himawari-8 | ViT-B | ✗ | 63.37 (−0.37) | 74.66 (−0.48) |

### 3.4. Ablation Study

#### 3.4.1. End-to-End Pre-Training

Integrating architectural ideas from the SFD model into MAE brings consistency between the pre-training architecture with fine-tuning architecture, which allows the whole knowledge learned by pre-training, i.e., pre-training weights, can be transferred to almost every corner of the fine-tuning network. In the SeaMAE experiments of Table 4, the fine-tuning results are obtained with such an end-to-end knowledge transfer by default. Therefore, Table 5 compares using only the encoder pre-trained weights with both encoder and decoder pre-trained weights, suggesting that the SFD's decoder initialization with pre-trained weights is a promising pre-training methodology.

**Table 5.** Comparison of loading SeaMAE pre-training weights to the decoder with randomly initializing the decoder. Green numbers are the performance gains derived from initializing the decoder with SeaMAE pre-training weights.

| Encoder Init. | Decoder Init. | Encoder | IoU | Acc. |
|:---:|:---:|:---:|:---:|:---:|
| MAE | Random | ViT-Ti | 58.63 | 70.47 |
| SeaMAE | Random | ViT-Ti | 59.16 | 70.94 |
| SeaMAE | SeaMAE | ViT-Ti | 59.37 (+0.21) | 70.97 (+0.03) |
| MAE | Random | ViT-S | 60.48 | 71.87 |
| SeaMAE | Random | ViT-S | 61.02 | 72.16 |
| SeaMAE | SeaMAE | ViT-S | 61.55 (+0.53) | 72.52 (+0.36) |
| MAE | Random | ViT-B | 62.92 | 73.91 |
| SeaMAE | Random | ViT-B | 63.43 | 74.25 |
| SeaMAE | SeaMAE | ViT-B | 63.74 (+0.31) | 74.66 (+0.41) |

#### 3.4.2. Data Scalability and Pre-Training Time

The standard MAE learns representations on ImageNet involving 1.43 million images. In contrast, our meteorological pre-training dataset contains 0.51 million images. Indeed, the meteorological MIM pre-training with one third of ImageNet's sample numbers leads to better SFD performance. We are interested in (1) how much performance drops if under some meteorological scenario where available data are insufficient, and (2) how much performance increases if under some meteorological scenario where more large-scale data are available.

For the first question, we reduce the pre-training dataset to 25% (0.125 m), 50% (0.25 m), and 70% (0.375 m) of total samples, respectively. Figure 6a draws the fine-tuning results of ViT at various sizes under four pre-training data scales and plots the trend curve. According to the trend curve, it can be observed that the fine-tuning performance is linearly correlated with the pre-training data scale.

For the second question, although it is possible to collect data before 2018 to enlarge the pre-training dataset, the large time span may result in a significant inner variance of representations in pre-training, which may impair performance. Thus, we adopt a robust alternative trick to simulate a larger-scale pre-training dataset. The epoch number is extended from 300 to 400, 600, and 800, approximately simulating a scaling factor of 1.33, 2, and 2.66 times the pre-training dataset. Figure 6a demonstrates this trend, showing that increasing the training epochs leads to better fine-tuning performance.

Based on these analyses, further increasing the data scale on SeaMAE is expected to yield performance gains in fog detection. Additionally, Figure 6b also examines the performance with reduced pre-training epochs. By comparing the results in Figure 6a with those in Figure 6b where the epochs are less than 300, we find that while varying the pre-training epochs can simulate data scaling, shorter learning schedule on a larger dataset induces higher performance, which also reflects the fast convergence of SeaMAE.

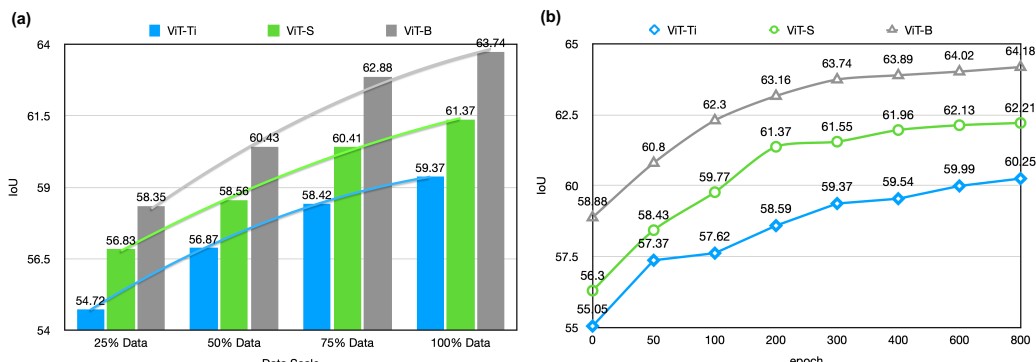

**Figure 6.** (**a**) Performance (IoU) change induced by scaling the pre-training data. (**b**) Performance (IoU) change induced by increasing the pre-training epochs.

### 3.4.3. Band Number of Input

The images captured by the Himawari-8 satellite are multi-channel images with 16 channels. In the conducted experiments both in pre-training and fine-tuning, all channels are input to the network. However, for the annotation of the fine-tuning dataset, we only labeled the fog based on the synthesized images from the three most prominent channels: 3, 4, and 14. Some previous literature has also focused on SFD research only using these three channels, despite that fine-tuning on all bands performs better than three channels [6,10]. To reduce the overhead of processing multi-spectral data, Table 6 explores two learning strategies where only three channels are used for fine-tuning. The first strategy uses only a combination of three channels, namely (3, 4, 14), for both pre-training and fine-tuning. The first line in the second group of results of Table 6 shows this way makes the performance degrades by 0.66. The second is using all bands to pre-train, and three bands to fine-tune. With this strategy, the input channel numbers of the input layer, i.e., patch embedding layer, in pre-training and fine-tuning are different, 16 and 3. So three methods are introduced, *Random*, *Index*, and *Resize*, to adjust the patch embedding to transfer the knowledge of the input layer. *Random* means the input layer in fine-tuning loads no knowledge, whose parameters are randomly initialized. *Index* means the input layer only loads the parameters of (3, 4, 14) channel. Denoting the weights of patch embedding layer from pre-training as $W_{pt} \in \mathbb{R}^{(ps \times ps \times 16 \times dim)}$, where $pt$ is short for pre-training, $ps$ is short for patch size, and $dim$ is short for dimension of ViT, *Index* makes the patch embedding layer in fine-tuning just load $W_{pt}[:, :, [3, 4, 14], :]$. *Resize* means $W_{pt}$ is resized to $W_{ft} \in \mathbb{R}^{(ps \times ps \times 3 \times dim)}$, which acts on the input channel through bicubic interpolation. Table 6 compares these three methods and shows that *Random* and *Index* both impair the performance apparently but *Resize* only causes an acceptable performance drop $0.22 IoU$. This very slight performance drop indeed improves the performance of SFD using only (3, 4, 14) bands to fine-tune, implying our pre-training approach is well adaptive in various band number settings.

**Table 6.** Comparison of different settings of band number in pre-training and fine-tuning in on ViT-B. **<span style="color:red">Red</span>** numbers are performance drops compared to using all bands both in pre-training and fine-tuning.

| Pre-Train Band# | Fine-Tune Band# | Patch Emb. Adjustment | IoU | Acc. |
|---|---|---|---|---|
| 16 | 16 | — | 63.74 | 74.66 |
| 3 | 3 | — | 63.08 (**−0.66**) | 73.99 (**−0.67**) |
| 16 | 3 | Random | 62.59 (**−1.15**) | 73.43 (**−1.23**) |
| 16 | 3 | Index | 62.71 (**−1.03**) | 73.50 (**−1.16**) |
| 16 | 3 | Resize | 63.52 (**−0.22**) | 74.48 (**−0.18**) |

### 3.5. Comparison to Other Satellite Imagery Pre-Training Methods

SatMAE [21] and RingMo [22] are two outstanding pre-training methods, both of which utilize MIM and learn representations on large-scale satellite imagery. What they employed as pre-training network architecture are improved versions of MAE. SatMAE introduces a temporal encoding for temporal imagery and a new masking strategy for input to MAE. RingMo also proposes a new masking strategy and a lightweight decoder with L1 loss for reconstruction. Table 7 shows the SFD performance using SatMAE and RingMo to learn representations of meteorological imagery and as the pre-training model. On backbone with different model capacities, SatMAE and RingMo show varying performance. Therefore, we calculate the performance difference between SeaMAE and the best-performing method among them, as shown in Table 5 for each group of results. SeaMAE outperforms the current state-of-the-art satellite imagery pre-training methods, yielding the best results in pre-training with meteorological satellite imagery for SFD.

**Table 7.** Comparison of SeaMAE pre-training methods proposed for satellite imagery. Results in **bold** font are the better ones between SatMAE and RingMo. **Green** numbers are the performance gains compared to **bold** results.

| Pre-Train Method | Pre-Train Data | Encoder | IoU | Acc. |
|---|---|---|---|---|
| SatMAE | Himawari-8 | ViT-Ti | **58.56** | **70.28** |
| RingMo | Himawari-8 | ViT-Ti | 58.22 | 69.90 |
| SeaMAE | Himawari-8 | ViT-Ti | 59.37 (**+0.81**) | 70.97 (**+0.69**) |
| SatMAE | Himawari-8 | ViT-S | **60.26** | 71.55 |
| RingMo | Himawari-8 | ViT-S | 60.14 | **71.61** |
| SeaMAE | Himawari-8 | ViT-S | 61.55 (**+1.29**) | 72.52 (**+0.91**) |
| SatMAE | Himawari-8 | ViT-B | 62.83 | 74.39 |
| RingMo | Himawari-8 | ViT-B | **63.02** | **74.42** |
| SeaMAE | Himawari-8 | ViT-B | 63.74 (**+0.72**) | 75.12 (**+0.70**) |

### 3.6. Qualitative Results of Sea Fog Detection

Figure 7 compares SFD qualitative results of Himawari-8 pre-training SeaMAE with ImageNet pre-training MAE. In terms of friendly visualization, the group of bands (3, 4, 14) is used to show the image. Looking into their SFD results, we find that ours presents more consistent annotations, more complete sea fog objects, and more precise boundaries between fog and background. For example, in the 3rd row of the first column, ImageNet pre-training one outputs two holes in the central area of sea fog, and mistakes a cluster of clouds as sea fog, but our result is very close to the ground truth with good prediction consistency.

Figure 7 also demonstrates that in SFD, the most challenging objects to differentiate from sea fog are various types of clouds, which can be categorized as high clouds and low clouds based on their altitude. Analyzing the visualization from bands (3, 4, 14) of meteorological satellite imagery, sea fog areas appear mostly smooth in texture, continuous in color, and dense in structure. In contrast, most low cloud areas show sparse structures, rough textures, and diverse color tones. However, since low cloud areas can also be extensive during sea fog occurrence, there are some sub-regions of low cloud where they resemble sea fog and exhibit similar representations. Experienced meteorologists can distinguish between them based on their rich experience. Nevertheless, for deep-learning models, it is still prone to making errors, especially False Positive predictions. For instance, in the 1st row and 2nd row of the 1st column, SeaMAE pre-training model shows more accurate predictions compared to ImageNet. Another challenge arises from the presence of high clouds frequently over a large area. High clouds are much sparser and rougher, and sometimes appear in a scattered distribution, unlike sea fog and low clouds that tend to cluster together. High clouds often present in a thin state and may overlay the sea fog, resulting in varying representations that can be easily overlooked by SFD models, leading

to False Negative predictions. For example, in the 4th row of the 1st column and the 3rd row of the 2nd column, SeaMAE accurately detects sea fog even in these overlaid circumstances, unlike ImageNet's incomplete predictions.

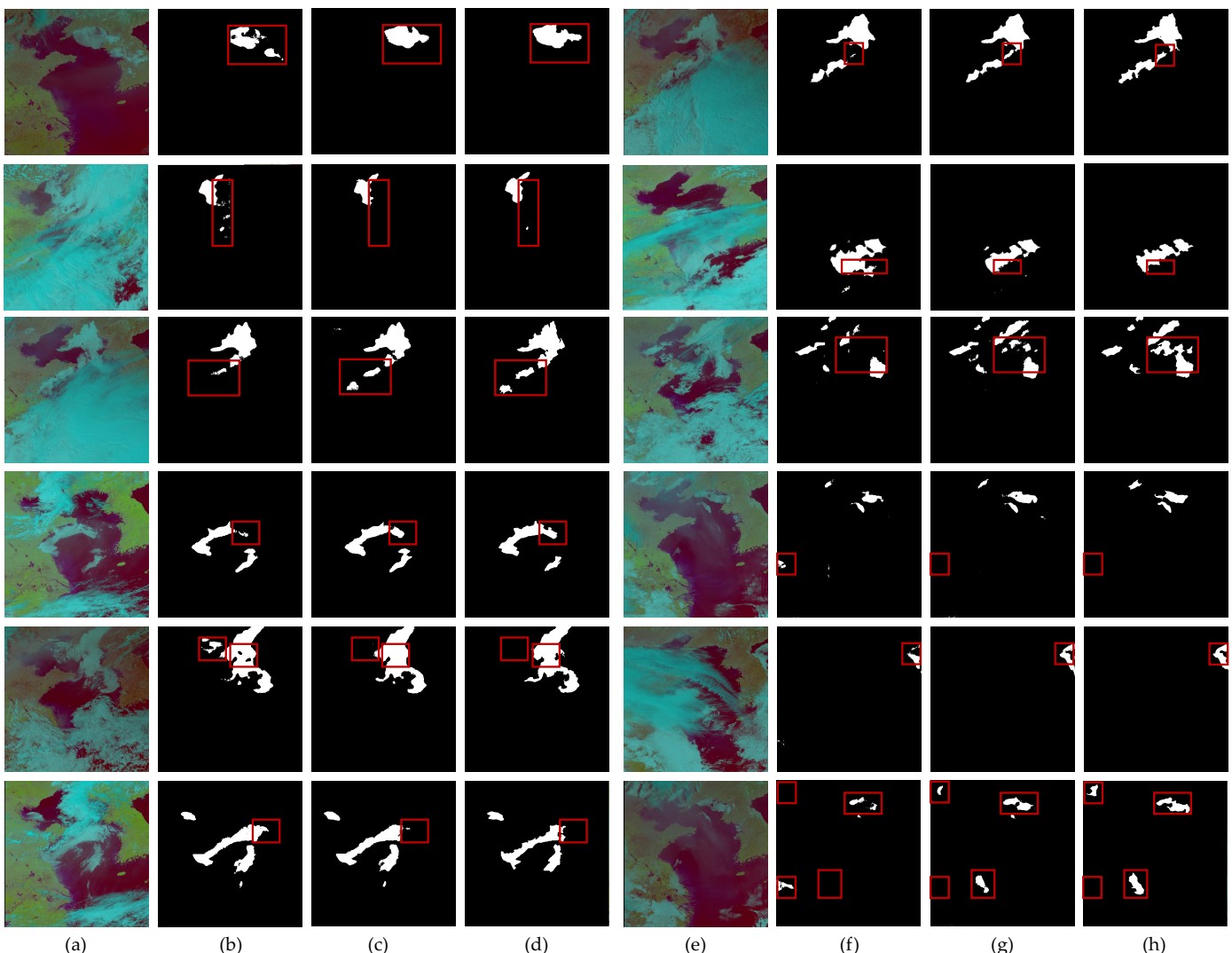

|          |          |          |          |          |          |          |          |
|:--------:|:--------:|:--------:|:--------:|:--------:|:--------:|:--------:|:--------:|
| (a)      | (b)      | (c)      | (d)      | (e)      | (f)      | (g)      | (h)      |

**Figure 7.** Comparison of SFD results . Red bounding boxes help to focus the reader's attention on the key area. (**a**,**e**) show the image. (**b**,**f**) show the ImageNet pre-training SFD results. (**c**,**g**) show the SeaMAE pre-training SFD results. (**d**,**h**) show the ground truth of sea fog.

### 3.7. Visualization

Qualitative Results of MIM

Figure 8 shows some reconstruction results. The network is pre-trained SeaMAE with ViT-B as the backbone, and we use the validation data in fine-tuning as input to evaluate its reconstruction performance qualitatively. Due to using all bands are not friendly to visualization, we just select the (3, 4, 14) band group to showcase. The objects in meteorological satellite imagery mainly are cloud and fog. SeaMAE succeeds in the reconstruction of the shape of main objects, even the very tiny and hazy ones. For detailed textures, SeaMAE can also depict them ambiguously but not that vividly because the cloud or fog has very sophisticated forms. In general, the pre-trained SeaMAE model demonstrates the ability to model images that capture representations of clouds and fogs, including their shape, size, region, and quantity, utilizing only a small portion of remaining patch cues.

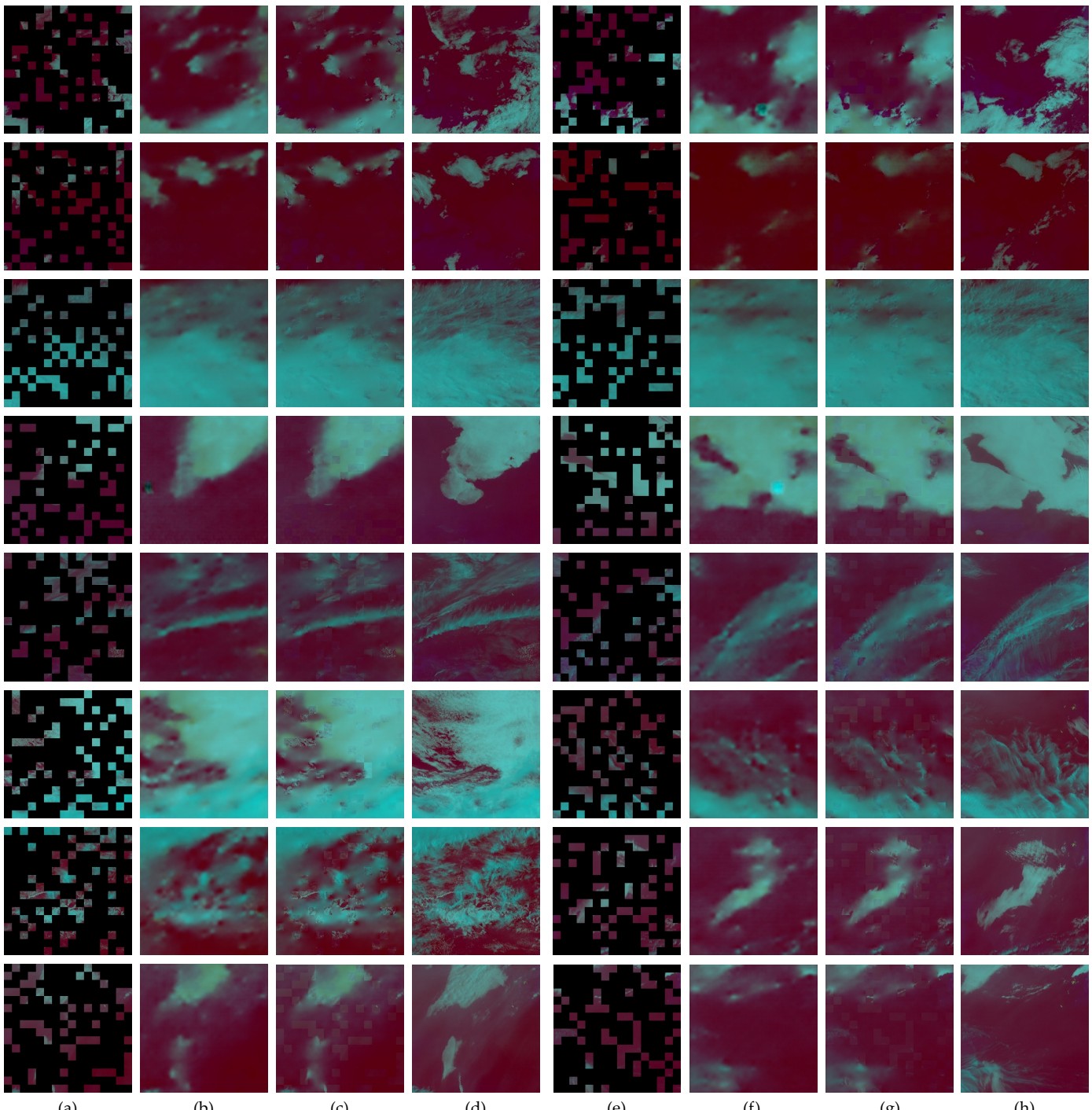

|  |  |  |  |  |  |  |  |
|---|---|---|---|---|---|---|---|
| (a) | (b) | (c) | (d) | (e) | (f) | (g) | (h) |

**Figure 8.** Reconstruction Results from SeaMAE based on ViT-B. (**a**,**e**) show the masked satellite image. (**b**,**f**) show the raw reconstruction output of SeaMAE. (**c**,**g**) show the combination of the reconstructed patch with the unmasked patch. (**d**,**h**) show the raw satellite image.

## 4. Discussions

The content of this discussion includes aspects: (1) Explanation of Results. (2) Validation over 2021 sea fog events. (3) Literature review and comparison to ours. (4) Future directions.

### 4.1. Explanation of Results

The initial step in applying deep-learning algorithms is to select a pre-training model and load its weights as the initialization of the deep network. Appropriate pre-training methods can lead to performance improvements and more stable optimization. Therefore, when using deep networks to interpret meteorological satellite imagery for sea fog detection (SFD), it is essential to utilize a pre-training model. Our results confirm that applying the most commonly used ImageNet pre-training model yields better results than training the SFD from scratch, i.e., with random parameter initialization.

However, ideally, the distribution of knowledge learned during pre-training should align with the knowledge that the deep-learning model needs to learn during the SFD learning phase (hereinafter referred to as fine-tuning). For instance, recent studies have shown that when interpreting very high-resolution (VHR) satellite imagery, pre-training the model with VHR satellite imagery (using self-supervised methods) outperforms pre-training on natural images (e.g., the ImageNet dataset), leading to better interpretation performance [21,22]. As both VHR satellite imagery and the Himawari-8 meteorological satellite imagery learned by the SFD model have some similarities, such as (1) bird's-eye camera angle and (2) densely arranged small objects, we compared the performance of the SFD model pre-trained on both natural imagery and VHR satellite imagery, and the results showed that the latter slightly outperformed the former, which meant closing the distribution gap between the pre-training data and the fine-tuning data is promising.

We continue to make efforts to bring the pre-training data representation closer to the fine-tuning data because there are still big gaps between VHR satellite imagery and meteorological satellite imagery, such as (1) channel numbers and types, (2) stationary background in SFD and non-stationary background in VHR satellite imagery, and (3) disparity of image resolution. To achieve this, we collected a large-scale pre-training dataset consisting of images captured by the Himawari-8 meteorological satellite with ample meteorologically marine representations, covering the same marine region and spectrum as the fine-tuning data for SFD. The meteorological pre-training approach largely outperformed the two previously mentioned pre-training methods and utilized only one third of their magnitudes. In the context of deep learning for SFD, our results revealed, for the first time, the importance of pre-training and pre-training data representation, indicating that pre-training the deep-learning model with meteorological satellite imagery can lead to marked SFD performance gains.

On the other hand, besides the data for pre-training, we delve into the network architecture for pre-training. By effectively leveraging Masked Image Modeling (MIM), masked autoencoders (MAE) [15] have emerged as state-of-the-art pre-training architecture. When comparing it with our designed SFD network architecture, it becomes evident that apart from the output layer, which differs due to the different prediction tasks (MIM and SFD), the main distinction lies in the decoder structure. Therefore, we hypothesized that the decoder structure of the SFD network (including hierarchical decoders and skip connections) may be inherently equipped with suitable inductive biases for processing meteorological satellite imagery (as introduced in Section 1), consequently, during the meteorological pre-training, if the decoder of MAE is replaced with the SFD network's decoder, it could result in further improvements in SFD. The results validated this hypothesis, as incorporating the architectural characteristics of the SFD network's decoder into the pre-training architecture MAE, be it hierarchical decoders or skip connections, further augmented SFD performance. In addition, the existing state-of-the-art endeavors on pre-training architecture for satellite imagery were compared to ours. The results clarify that for SFD with meteorological satellite imagery, our architecture is the best for pre-training.

Furthermore, we uncovered another advantage of such architectural improvement, which is the complete utilization of the pre-training model. Typically, due to the differences (learning targets) between pre-training and fine-tuning, the decoder structures of the pre-training and fine-tuning models also differ. As a result, the utilization of the pre-training model involves loading only the encoder's pre-training weights. However, our improvement keeps the consistency of decoders between pre-training and fine-tuning, enabling the complete utilization of the pre-training model, not only loading the encoder's weights but also the decoder's to the SFD network. The results further analyzed the contributions of such architectural improvement in pre-training, revealing that the complete utilization of the pre-training model indeed elevates SFD performance.

Although our efforts in pre-training, whether in terms of data or network architecture, were initially aimed at improving SFD's performance, the analysis of results motivates us to assume that the proposed method has generalizability for most meteorological satellite imagery interpretation tasks, especially in marine regions. In summary, collecting a large-scale dataset of meteorological satellite imagery that resembles the downstream learning task's regions, meteorological activities, and satellite characteristics, and employing a decoder structure that exhibits excellent performance in that task as the pre-training network's decoder, holds promising potential as a pre-training methodology.

### 4.2. Validation over 2021 Sea Fog Events

The validation dataset involves all sea fog events in 2021. Table 8 reports, in 2021, when the sea fog event happened and how long time the sea fog event lasted. Table 9 presents monthly statistics of the number of sea fog events. From these two tables, it can be observed that sea fog is most frequent during the spring season, with no occurrences during the autumn season. Sea fog also occurs during the summer and winter seasons, but the total number of occurrences is lower compared to the spring. In general, the duration of sea fog events lasts around 7 to 8 h, but in July, there were two events with much shorter durations, due to the influence of tropical cyclones during the summer.

Table 8 further compares the performance of the SFD, where the backbone is ViT-Base with the ImageNet and SeaMAE pre-training weights for each sea fog event. Our method outperforms the SFD performance from the ImageNet pre-training weights on any sea fog event. This indicates that the proposed pre-training paradigm is not influenced by specific sea fog events, occurrence dates, or seasons, and consistently presents performance gains for sea fog detection.

**Table 8.** Sea fog event records in 2021. : Date, Duration, ImageNet pre-training SFD IoU, and SeaMAE pre-training SFD IoU from top to bottom in every cell. Green numbers are performance gains derived from using the SeaMAE pre-training weights instead of ImageNet pre-training ones.

| Jan. 21st | Jan. 24th | Jan. 26th | Feb. 11th | Feb. 12th | Feb. 13th |
|---|---|---|---|---|---|
| 8 h | 8 h | 8 h | 8 h | 8 h | 8 h |
| 49.69 | 40.52 | 61.70 | 76.51 | 61.32 | 46.43 |
| 74.28 (+24.59) | 64.86 (+24.34) | 72.75 (+11.05) | 77.63 (+1.12) | 63.93 (+2.61) | 59.59 (+13.16) |
| **Feb. 20th** | **Mar. 5th** | **Mar. 10th** | **Mar. 14th** | **Mar. 25th** | **Mar. 28th** |
| 8 h | 8 h | 8 h | 8 h | 8 h | 8 h |
| 54.05 | 58.34 | 58.77 | 56.37 | 90.55 | 56.10 |
| 63.08 (+9.03) | 76.61 (+18.27) | 60.42 (+1.64) | 58.86 (+2.49) | 91.12 (+0.57) | 68.03 (+11.93) |
| **Apr. 26th** | **Apr. 27th** | **Apr. 28th** | **Apr. 29th** | **May 9th** | **May 10th** |
| 7.5 h | 8 h | 6.5 h | 6.5 h | 8 h | 8 h |
| 64.67 | 55.81 | 59.36 | 58.07 | 87.94 | 72.94 |
| 65.28 (+0.61) | 59.64 (+3.83) | 64.11 (+4.75) | 62.43 (+4.36) | 88.08 (+0.14) | 73.86 (+0.92) |
| **May 30th** | **May 31st** | **Jun. 6th** | **Jun. 13th** | **Jul. 11th** | **Jul. 17th** |
| 8 h | 8 h | 8 h | 8 h | 3 h | 5.5 h |
| 68.73 | 75.95 | 79.03 | 80.00 | 49.61 | 45.99 |
| 70.41 (+1.68) | 77.81 (+1.86) | 80.13 (+1.10) | 80.34 (+0.34) | 64.96 (+15.35) | 68.44 (+22.45) |

**Table 9.** Number of sea fog events in every season and every month.

| Winter | | Spring | | | Summer | |
|---|---|---|---|---|---|---|
| Jan. | Feb. | Mar. | Apr. | May | Jun. | Jul. |
| 3 | 3 | 5 | 4 | 4 | 2 | 2 |

During a sea fog event, the position and shape of the sea fog gradually change over time. Although Table 8 calculates the Intersection over Union (IoU) for each sea fog event as a criterion of performance, we aim to further highlight the superiority of our proposed pre-training paradigm for SFD intuitively. To achieve this, we devise high-frequency SFD extraction along the time dimension over the entire duration of one event. This strategy allows for summarizing the sea fog event using a single sea fog mask that captures the climactic sea fog (CSF) locations throughout the entire sea fog event.

For a specific location denoted by pixel $(i, j)$ in the study region, if sea fog occurs at that location for more than half of the time steps within the sea fog event duration $(0, T)$, the pixel $(i, j)$ is defined as a CSF location. Both ground truth sea fog events and predicted sea fog events can be processed to obtain CSF locations in this way, resulting in the constitution of CSF masks.

Figure 9 compares the CSF masks predicted by the ImageNet pre-training model, the SeaMAE pre-training model, and the ground truth sea fog events. The CSF masks of predictions are contrasted with the ground truth CSF masks. The yellow regions represent True Positive predictions, the red regions represent False Negative predictions, and the orange regions represent False Positive predictions. Across all sea fog events, our proposed paradigm consistently generates predictions that most closely resemble the ground truth CSF masks.

*4.3. Literature Review and Comparison to Ours*

In comparison with related literature, this work's contributions and significance can be highlighted.

4.3.1. Sea Fog Detection in Bohai Sea and Yellow Sea

We primarily review works that use deep-learning technology to detect sea fog (SFD) [3–14]. In recent years, most of them studied the area covering the Bohai Sea and Yellow Sea [3,6,10–14], where the sea fog is a frequent occurrence. Ref. [3] introduces a scSE-LinkNet model that utilizes residual blocks and an attention module to accurately detect daytime sea fog. Ref. [6] presents a correlation context-driven method for SFD using a two-stage superpixel-based fully convolutional network (SFCNet) and an attentive Generative Adversarial Network (GAN). Ref. [10] introduces an unsupervised domain adaptation method that utilizes labeled land fog data to detect sea fog in meteorological satellite imagery. Ref. [11] presents an improved algorithm using one visible (VIS) and one near-infrared (NIR) band of the Himawari-8 satellite, demonstrating successful SFD without cloud mask information. Ref. [12] proposes a two-stage deep-learning strategy for daytime SFD through the creation of a labeled dataset and the Cloud-Aerosol LiDAR with Orthogonal Polarization (CALIOP) vertical feature mask (VFM). Ref. [13] utilizes MODIS images to analyze the temporal and spatial characteristics of sea fog, establishing a threshold algorithm based on specific MODIS bands. Ref. [14] presents a novel approach to weakly supervised semantic segmentation for SFD using point-based annotation and auxiliary information from ICOADS visibility data.

Previous methods oftentimes employ ImageNet pre-training models or learn from scratch on meteorological satellite imagery annotated with sea fog. In contrast, our novel learning paradigm prioritizes large-scale pre-learning with meteorological satellite imagery, capitalizing on the rich availability of multi-spectral geostationary imagery captured by the meteorological satellite. Furthermore, we strengthen this paradigm by harnessing and amending Masked Autoencoders (MAE), for the first time enabling the integration of Vision Transformers into SFD, a pioneering endeavor within this interdisciplinary community.

Moreover, our approach extends the scope of pre-training beyond the encoder, embracing the decoder of the SFD network as an essential pre-trained component.

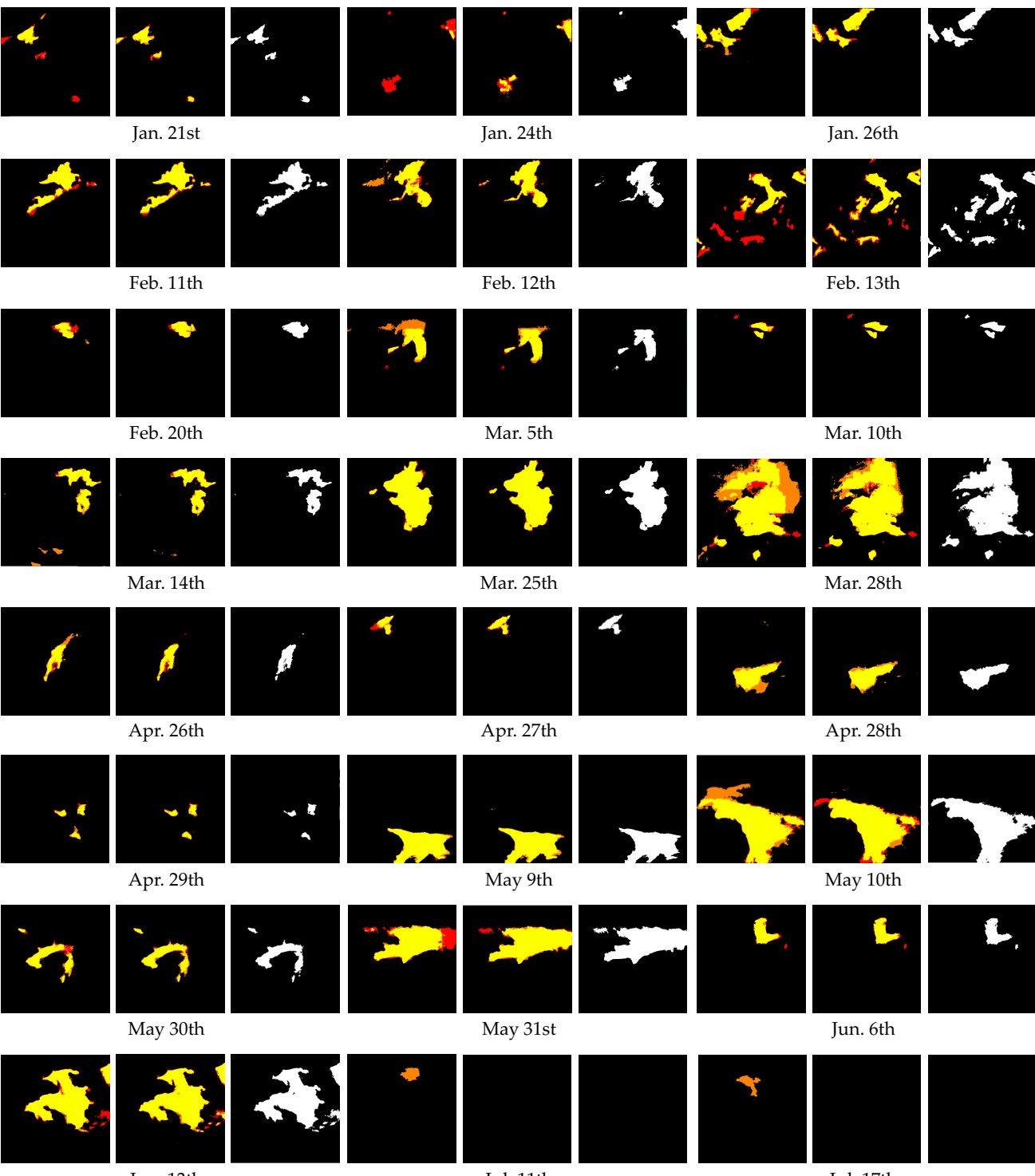

**Figure 9.** CSF masks from ImageNet pre-training prediction , SeaMAE pre-training prediction, and GT in every sea fog event involved within the validation dataset. **Yellow** pixels represent True Positive predictions. **Orange** pixels represent False Positive predictions. **Red** pixels represent False Negative predictions. Zoom in to View.

4.3.2. MIM Pre-Training with Satellite Imagery

Recently MIM succeeds in pre-training very large-scale Vision Transformers [15] in many fields [19,20,34–37], so more and more Vision Transformers pre-trained with satellite imagery have been being proposed [21,22,38–56]. Ref. [38] introduces a scalable and adaptive self-supervised Transformer (AST) model for optical satellite imagery interpretation by generating scale-invariant representations through cross-scale collaboration and masking strategies. Ref. [39] presents large vision models specifically designed for VHR satellite imagery interpretation, utilizing rotated varied-size window attention in transformers. Ref. [40] introduces Presto, a pre-trained Remote Sensing Transformer specifically designed for self-supervised learning on remote-sensing pixel-time series data. Ref. [22] introduces RingMo, a remote-sensing foundation model framework that leverages generative self-supervised learning to construct a large-scale dataset—Million-AID—and train models with improved general representation capabilities. Ref. [21] introduces SatMAE, a pre-training framework for satellite imagery based on Masked Autoencoder, leveraging temporal and multi-spectral information. Ref. [41] explores the use of self-supervised MIM and modifications to the transformer architecture for hyper-spectral satellite imagery. Ref. [42] proposes RS-BYOL, a self-supervised learning approach for remote sensing, which leverages multi-spectral and synthetic aperture radar data as an implicit augmentation to learn invariant feature embeddings.

Most of the existing studies in the field of MIM pre-training with satellite imagery use large-scale satellite datasets to pre-train models, tailored to the downstream task of interpreting VHR satellite imagery. These pre-training datasets typically involve various geographical scenarios and various types of satellite imagery, such as multi-high-resolution, multi-spectral, optical RGB, and SAR. However, the marine environment and meteorological satellite imagery have remained unexplored in these works. In comparison to the existing literature, SeaMAE serves as the first contribution in this community to intelligent vision tasks of ocean observation, i.e., SFD, through a novel MAE architecture with architecture characteristics from the SFD network and a large-scale pre-training dataset consisting of meteorological satellite imagery.

*4.4. Future Directions*

Our future research will consist of a probe into the used fine-tuning dataset, a validation of the deduction proposed at the end of Section 4.1, which will be carried on another challenging interpretation of meteorological imagery, cloud classification in pixel, and an extended application on pre-training SFD deep-learning models in other marine regions.

The SFD dataset used in our study is finely annotated by experts with vast knowledge of sea fog, covering all sea fog events in the Bohai Sea and Yellow Sea over three years, from 2019 to 2021. Considering the scarcity of satisfactory SFD datasets in the community, we will make this dataset publicly available. Apart from benchmarking SFD technology, this dataset can also be utilized for research on cloud classification in meteorological satellite imagery, therefore promoting the intelligent development of marine meteorology. Encyclopedic statistics of the dataset will be analyzed in a separate article, and we will propose a novel SFD method capable of setting state-of-the-art results on this dataset.

In Section 4.1, we draw a deduction that the methodology proposed in this paper is likely to apply to multiple meteorological satellite imagery interpretation tasks, such as pixel-level cloud classification in marine regions. From the perspective of pixel-level classification, i.e., semantic segmentation or scene understanding, SFD is a binary classification, distinguishing pixels as either sea fog or background. On the other hand, cloud classification involves categorizing pixels into ten classes, including alto-cumulus, altostratus, nimbostratus, cumulus, stratocumulus, stratus, cirrus, cirrostratus, and deep convection. We expect to revisit pixel-level cloud classification in meteorological satellite imagery, building upon the pre-training guideline proposed in this paper. Such research will be to explore a pre-training paradigm tailored for cloud classification. Additionally, the deduction drawn

from the above results in pre-training for SFD can be examined by the study on cloud classification, one of the most important meteorological imagery interpretation tasks.

In future work, validating the proposed learning paradigm for SFD in other marine regions, or devising specialized pre-training strategies, will both significantly advance the field. We will also proceed in this direction by conducting SFD research in the South China Sea and integrating it with the Bohai Sea and Yellow Sea regions to propose a highly generalized pre-training model tailored for SFD across multiple marine areas.

## 5. Conclusions

This paper develops a novel learning paradigm for sea fog detection (SFD). We focus on an economically important and sea-fog-vulnerable area in the range of 115.00°E to 128.75°E and 27.60°N to 41.35°N. Our learning paradigm improves the SFD performance of canonical learning paradigms by 2–3% sea fog IoU, achieving 64.18% sea fog IoU. We primarily answer the question that how to pre-train SFD networks. This paper is also the first research utilizing Vision Transformer as the backbone of SFD networks. First, we show that ViT without pre-training will lead to unsatisfactory performance, and pre-training in any manner is of significance. Next, we show that ImageNet pre-training and VHR satellite imagery pre-training can be substituted by the meteorological satellite imagery pre-training. Finally, we show that making the pre-training network architecturally consistent with the sea fog detection model can brings more performance gains, which is mainly attributed to (1) the architectural characteristics of the SFD model are also effective in meteorological sea fog reconstruction and (2) such an architectural consistency enables the decoder part of SFD model to load pre-training knowledge. Even when compared to the latest pre-training methods specialized for satellite imagery, our approach still demonstrates considerable superiority in meteorological satellite imagery pre-training for SFD. We hope this work can inspire creativity in pre-training deep-learning models for interpreting meteorological satellite imagery in the future.

Limitations: This paper investigates the pre-training of deep-learning models for SFD in meteorological satellite imagery. However, due to the availability of large-scale data (more sea fog events in the Bohai Sea and Yellow Sea region), our analysis is limited to SFD in this specific area. Therefore, whether the proposed approach is effective for SFD in other sea regions deserves further investigation.

**Author Contributions:** Conceptualization, H.Y. and Y.Z.; Methodology, H.Y. and Y.Z.; Software, S.S. and Y.Z.; Validation, S.S. and M.X.; Formal analysis, H.Y.; Investigation, H.Y.; Resources, S.S.; Data curation, S.S. and M.X.; Writing—original draft, H.Y.; Writing—review & editing, M.W. and C.Z.; Visualization, S.S.; Supervision, M.W., C.Z. and B.H.; Project administration, M.W., M.X., C.Z. and B.H. All authors have read and agreed to the published version of the manuscript.

**Funding:** This work is supported by National Key R&D Program of China No. 2021YFC3000905.

**Data Availability Statement:** The data used in this study is not currently available. But we will make it available in a few months' time as we continue to work on it. Anyone who needs the data can contact us by e-mail for latest information.

**Conflicts of Interest:** The funders had no role in the design of the study; in the collection, analyses, or interpretation of data; in the writing of the manuscript; or in the decision to publish the results.

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
