# Peer review of "SeaMAE: Masked Pre-Training with Meteorological Satellite Imagery for Sea Fog Detection"

_remotesensing, doi:10.3390/rs15164102_

Round 1
Reviewer 1 Report
SeaMAE: Masked Pre-training with Meteorological Satellite Imagery for Sea Fog Detection
This paper uses a deep learning method to sea fog detection driven by masked image modeling. The study shows that pre-training with meteorological satellite imagery can improve the sea fog detection compared to pre-training with nature imagery and VHR satellite imagery, and incorporating the architectural characteristics of sea fog detection models into a vanilla masked autoencoder can augment the effectiveness of meteorological pre-training. This study may be of interest to those concerned about sea fog with space-based observations though the improved accuracy is not very significant. The article is well written and looks clear and professional, thus I suggest a little addition and clarification before publication.
Comment:
1. My main concern is whether the authors can do some work on improving the model algorithm or the sea fog detection model.
2. line 325, check ‘Figure ??’
3. Table 5, ‘finetuning’ → fine-tuning
4. Figure 6, ‘SeaMae’ → SeaMAE
The article is well written and looks clear and professional but with a few typing errors.
Reviewer 2 Report
Sea fog detection (SFD) study has potential application value for oceanic remote sensing. This manuscript proposed a machine learning method SeaMAE for a better accuracy of sear fog detection by defining a new MAE using vision Transformer as the encoder and a convolutional hierarchical decoder. I recommend to reject this manuscript as following reasons: 1 The paper was organized and written irregularly. The structure of introduction, method and discussion need to be improved. Such as what is the accuracy of existing methods of SFD should be shown in the introduction chapter. And the last part of introduction which describes the ideas and advantages of the proposed SeaMAE method should be in "Method" and "Conclusion" chapters. 2 The validation is not enough. Comparison with previous studies should be carried out contrastively. Such as the accuracy, the wrongly detected pixels in your method and others and why, and the performances on different regions, seasons and cloud types. 3 The final accuracy should be involved in the abstract and conclusion. 4 The dataset was not introduced sufficiently and clearly.Author Response
Please see the attachment.

Reviewer 3 Report
The paper's contributions can be summarized into two key points. First, it proposes a novel learning paradigm for sea fog detection (SFD) that improves the performance of existing paradigms by 2%-3% in terms of sea fog IoU. Second, it introduces the use of Vision Transformer as the backbone of SFD networks
In Figure 4.c, the proposed learner paradigm is depicted. However, it is unclear how the learner makes decisions to detect sea fog and reconstruct the imagery using the same decoder. There seems to be a misleading aspect in the block diagram or the overall learner design. Further discussion or illustration is necessary to clarify this aspect. The authors are encouraged to provide a more detailed explanation of the decision-making process and the integration of detection and reconstruction in the learner paradigm
Limited geographical scope: The paper focuses on a specific economically important and sea-fog vulnerable area within a narrow range of coordinates (115.00E to 128.75E and 27.60N to 41.35N). The conclusions drawn from this limited region may not be applicable to other geographical areas, potentially limiting the generalizability of the proposed learning paradigm.
Pre-training methods: While the conclusion asserts that pre-training in any manner is significant, it does not thoroughly evaluate or compare different pre-training methods. The paper only considers two potentially powerful self-supervised pre-training methods (ImageNet pre-training and VHR satellite imagery pre-training) and substitutes them with meteorological satellite imagery pre-training. However, the rationale behind this substitution and its implications on the performance of the SFD model are not adequately discussed or supported by experimental evidence
Lack of future directions: Although the conclusion expresses hope that the work will inspire future research on pre-training sea fog detection models, it does not provide specific suggestions or recommendations for further investigations. Identifying potential avenues for improvement or addressing the limitations of the proposed learning paradigm would enhance the usefulness of the paper for the scientific community.
